# New Insights into Mechanisms of Ferroptosis Associated with Immune Infiltration in Neonatal Hypoxic-Ischemic Brain Damage

**DOI:** 10.3390/cells11233778

**Published:** 2022-11-25

**Authors:** Shangbin Li, Li Wan, Jingfei Sun, Weichen Yan, Jie Wang, Xiong Gao, Changjun Ren, Ling Hao

**Affiliations:** 1Department of Pediatrics, First Affiliated Hospital of Hebei Medical University, Hebei Medical University, Shijiazhuang 050000, China; 2Institute for Epidemic Disease Control, Shijiazhuang Center for Disease Control and Prevention, Shijiazhuang 050000, China; 3Department of Pediatrics, Zhengding People’s Hospital, Shijiazhuang 050000, China

**Keywords:** hypoxic-ischemic brain damage, ferroptosis, immune infiltration, bioinformatics, newborn

## Abstract

Background: The mechanisms underlying ferroptosis in neonatal hypoxic-ischemic brain damage (HIBD) remain unclear. Method: Four microarray datasets were collected from the GEO database (three mRNA datasets GSE23317, GSE144456, and GSE112137, and one miRNA microarray dataset GSE184939). Weighted gene co-expression network analysis (WGCNA) was used to identify modules of HIBD-related genes. The ferroptosis-related genes were extracted from FerrDb, of which closely correlated to HIBD were obtained after the intersection with existing HIBD’s DEGs. Gene Ontology (GO) and Kyoto Encyclopedia of Genes and Genomes (KEGG) pathway enrichment analysis, as well as protein–protein interaction (PPI) network analysis were subsequently conducted. Cytoscape was used to identify central genes. Immune cell infiltration analysis was performed by the CIBERSORT algorithm. Result: Fifty-six ferroptosis-related differentially expressed genes (FRDEGs) were screened, mainly related to ferroptosis, autophagy, hypoxia response, metabolic pathways, and immune inflammation. The seven optimal hub FRDEGs were obtained by intersecting with key modules of WGCNA. Then, the expression levels of the seven optimal hub FRDEGs were validated in the GSE144456 and GSE112137 datasets, and the ferroptosis-related mRNA-miRNA network was established. In addition, this study revealed immune cell infiltration in the HIBD cerebral cortex and the interaction between immune cells. Moreover, notably, specific FRDEGs were strongly positively correlated with immune function. Conclusions: The mechanism of ferroptosis is intricate and closely related to neonatal HIBD. Therefore, targeting ferroptosis-related gene therapy and immunotherapy may have therapeutic prospects for neonatal HIBD.

## 1. Introduction

As one of the leading causes of neonatal death and neurological disorders, neonatal hypoxic-ischemic brain damage (HIBD) refers to the hypoxic-ischemic brain damage caused by perinatal asphyxia, with global incidence of 1–8 per 1000 live births approximately [1,2]. However, there is currently no specific treatment for HIBD in clinical practice. Despite the fact that mild hypothermia therapy is administered promptly, some neonates with severe perinatal asphyxia may develop serious complications, which brings a heavy burden to global public health [3,4]. Therefore, an effective treatment method for HIBD is urgently required.

Neuronal cell death after HIBD is the main cause of long-term neurological disorders. A growing number of studies have shown that different forms of cell death occur concurrently or sequentially [5]. Substantial evidence demonstrates that, after cerebral hypoxia and ischemia, the formation of reactive oxygen species (ROS) increases rapidly, directly modifies or degrades cellular macromolecules, including cell membranes, lipids, and DNA, which further triggers a cascade inflammatory response and the induction of oxidative stress [6]. Ferroptosis is a non-apoptotic, peroxidation-driven form of regulated cell death discovered in 2012 [7]. In terms of morphology, biochemistry, and genetics, it is distinct from apoptosis, necrosis, autophagy, and other forms of cell death. From a mechanistic perspective, unsaturated fatty acids highly expressed on cell membranes are catalyzed to produce lipid peroxidation, which induces cell death under the effect of divalent iron or ester oxygenase. Meanwhile, the core enzyme GPX4, which regulates the antioxidant system (glutathione system), is decreased [8]. A series of studies have shown that ferroptosis is closely related to the pathophysiological process of various brain diseases, such as ischemic and hemorrhagic stroke, Alzheimer’s disease, and Parkinson’s disease [9,10]. Studies have found that the dynamic changes of iron metabolism in different brain regions (parietal cortex, corpus callosum, hippocampus) in neonatal rats within 84 days after hypoxia-ischemia may cause lipid peroxidation to damage brain tissue [11]. This property suggests that there is a relationship between iron metabolism and tissue damage in HIBD. Recent studies have demonstrated that abnormal regulation of iron metabolism, amino acid metabolism, and lipid peroxidation in HIBD leads to the decreased antioxidant capacity of neurons and mitochondrial damage, leading to ferroptosis of neurons in the cerebral cortex [12]. Moreover, toll like receptor 4 activation promotes neuronal ferroptosis, and its inhibition can improve neuroinflammation [13]. Further, studies have confirmed that neuronal ferroptosis was observed in the neonatal HIBD rat model, and the injection of ferroptosis inhibitor melatonin can significantly inhibit neuronal ferroptosis, promote the survival of hippocampal neurons, and improve learning and memory abilities in HIBD rats [14]. In addition, the infiltration of peripheral immune cells can affect the tissue damage, protection, repair, and regeneration of HIBD [15]. More importantly, monitoring brain immunity and inflammation can guide clinical decision-making [16]. Ferroptosis is crucial in regulating the function of immune cells, mainly including the number and function of immune cells affected by ferroptosis of immune cells and the inflammatory immune response initiated by ferroptosis of non-immune cells [17]. However, the role and signaling pathway of ferroptosis in the pathogenesis of HIBD remains elusive. Therefore, the exploration of the potential mechanisms of ferroptosis and immune cell infiltration leading to HIBD cell death may provide new strategies and approaches for the prevention and treatment of neonatal HIBD.

Evidence has demonstrated that miRNAs are involved in the pathogenesis of neonatal HIBD [18,19]. Specifically, miR-30b can regulate plasminogen activator inhibitor-1(PAI-1), which in turn regulates HIBD pathogenesis and immune response [20]. Moreover, miR-384 has high diagnostic accuracy in differentiating neonates with HIBD of different severity [21]. Nevertheless, the regulatory relationship between miRNAs and neonatal ferroptosis in HIBD remains ambiguous mainly due to limited research [18].

With the advances in genomic microarray and high-throughput sequencing technologies, bioinformatics analysis is growing in popularity as a tool for exploring brain damage-related biomarkers and molecular mechanisms. As a novel invented systematic bioinformatic approach, WGCNA has been used to select co-expression modules and key modules, and to evaluate the correlations of gene modules with different clinical features, which provides a new means for exploring potential molecular mechanisms of disease [22].

Therefore, the purpose of this study is to identify the differential expression genes associated with ferroptosis in HIBD based on bioinformatics technology, to reveal the biological function and signaling pathway in which they participate, and to explore the correlation between ferroptosis-related genes and immune cell infiltration. The workflow is shown in Figure 1. Our results will help to shed light on new thoughts in the mechanism of neonatal HIBD.

## 2. Materials and Methods

### 2.1. Data Acquisition and Processing

Three mRNA datasets (GSE23317, GSE144456, and GSE112137) and one miRNA microarray dataset (GSE184939) were retrieved from the Gene Expression Omnibus (GEO) database (http://www.ncbi.nlm.nih.gov/geo/, accessed on 20 April 2022). Among them, we adopt dataset GSE23317 for analysis, while dataset GSE14456 and GSE112137 for validation. Expression matrix data of cerebral cortex samples of 22 8-day-old mice (P8) were obtained from GSE23317, including 3 time points (3 h, 8 h, and 24 h) of the cerebral cortex of HIBD mice and sham-operation mice, respectively. According to the design of Dupré et al. [23], in GSE144456, mice aged 10 days (P10) and 5 days (P5) mimic the condition of term infant HIBD and premature infant HIBD, respectively. Their experiment included the gene expression matrix data of the cerebral cortex tissue of 30 HIBD and sham-operated mice with different ages at 4-time points (3 h, 6 h, 12 h, and 24 h). According to the research of Paşca et al. [24], GSE112137 contains gene expression data after 24 h and 48 h of hypoxia treatment on human cortical spheroids (hCS, differentiated from human induced pluripotent stem cells) that transcriptionally resembled the cerebral cortex at 19–24 PCW. Meanwhile, microRNA data from the cerebral cortex of 4 HIBD models aged 9 days (P9) and 4 sham mice are included in the GSE184939 data file. The characteristics of all datasets are listed in Table 1. During this stage, we standardized datasets and annotated probes in them via the corresponding GEO platform (GPL). In particular, the mean value is used to indicate the expression level of the gene when multiple probes correspond to it.

### 2.2. Variation Analysis of mRNA and miRNA

Differentially expressed genes (DEGs) in the GSE23317 dataset and differentially expressed miRNAs (DEmiRNAs) in the GSE184939 dataset were screened via the limma package [25] in R software (version 4.0.5, Auckland, New Zealand). The filtering conditions were set to *p*-value < 0.05 and the absolute value of log fold change |log2FC| ≥ 0.3.

### 2.3. Ferroptosis-Related Gene Acquisition

FerrDb is a manually curated database for ferroptosis regulators and ferroptosis-disease associations from published journal articles [26]. The ferroptosis-related genes (FRGs) data files were screened from the FerrDb (http://www.zhounan.org/ferrdb/, accessed on 20 April 2022) based on the following criteria: 1. Knock out genes that are not expressed in humans and mice; 2. Knock out non-coding RNA. Afterward, ferroptosis-related DEGs (FRDEGs) were obtained by intersecting the screening results with DEGs.

### 2.4. Functional Enrichment and Signaling Pathway Analysis

DAVID 6.8 (https://david.ncifcrf.gov/home.jsp, accessed on 20 April 2022) is a powerful gene annotation tool that can perform Gene Ontology (GO) and Kyoto Encyclopedia of Genes and Genomes (KEGG) pathway enrichment analysis. In this study, the functions and signaling pathways of FRDEGs in the cerebral cortex of HIBD were annotated via DAVID 6.8. Particularly, *p* < 0.05 was used to judge the specificity of significant biological functions and signaling pathways.

### 2.5. Weighted Gene Co-Expression Network Analysis (WGCNA)

To further explore the potential role of DEGs in the occurrence and development of HIBD, the genes in the top 75% of the median absolute deviation in GSE23317 dataset were selected for WGCNA analysis [27]. R^2^ = 0.85 was set to filter the best soft threshold, while co-expression modules and key genes related to HIBD were obtained. Subsequently, the adjacency matrix is transformed into a topological overlap matrix (TOM), and modules were identified via hierarchical clustering (minModuleSize = 30). Specifically, eigengenes and modules are respectively calculated and hierarchically clustered. Thereinto, module eigengenes (ME) and module members (MM) can identify essential modules related to HIBD. Finally, the key modules can be obtained by computing the relationship between the representation data and the module. Thereinto, ME displays the first principal component in a module and describes the expression pattern of the module; MM represents the relationship between genes and module eigengenes.

### 2.6. Construction of Protein-Protein Interaction (PPI) Network

The protein–protein interaction (PPI) network is the process by which proteins form protein complexes through non-covalent bonds. The FRDEGs were imported into the STRING database (http://string-db.org/, accessed on 22 April 2022) to analyze their interactions. Statistically, a confidence scoring ≥ 0.4 was considered significant. Therefore, a PPI network graph is retrieved according to this criterion. Cytoscape (version 3.7.1, San Diego, CA, USA) is an open-source software for bioinformatics analysis, commonly used to visualize molecular interaction networks, while NetworkAnalyzer is a plugin for Cytoscape, which is used to perform topology analysis [28]. Results from the STRING database were imported into Cytoscape and hub genes (PPI-hub DEGs) were screened using degree value ≥ 3 in cytoHubba.

### 2.7. Identification and Validation of the Optimal Hub FRDEGs

In order to identify hub FRDEGs closely related to HIBD, the key genes identified by WGCNA (WGCNA-hub genes) and PPI-hub DEGs were intersected to obtain overlapping DEGs. These genes were defined as the optimal hub FRDEGs and verified in GSE144456 and GSE112137 datasets. Gene expression data were presented as mean ± standard deviation (SD). The differences in gene expression between the HIBD and control groups in different periods were compared via Tukey’s multiple comparisons test in GraphPad (Ver. 8.0.1, Chicago, IL, USA) software. Statistically, *p* < 0.05 was considered significant. Furthermore, the correlation of hub FRDEGs expression patterns was analyzed by a Pearson method. Last, but not least, compared with the preterm brain, whether the hub FRDEGs are expressed in human term brain was further determined via interrogating developmental transcriptome of the BrainSpan Atlas (brainspan.org, accessed on 10 November 2022).

### 2.8. Construction of Ferroptosis-Related mRNA-miRNA Regulatory Network

An mRNA-miRNA regulatory network was constructed to reveal the regulatory mechanisms of the optimal hub FRDEGs. The miRWalk3.0 online tool (http://mirwalk.umm.uni-heidelberg.de/, accessed on 25 April 2022) is a comprehensive miRNA target gene database, including the miRNA target genes information in multiple species, such as rats and humans [29]. First, miRNAs that may be involved in the regulation of the optimal hub FRDEGs were predicted via miRWalk. Then, they intersected with the DEmiRNAs of GSE184939 to obtain overlapping miRNAs. Finally, the miRNA-hub FRDEGs interaction network was constructed via Cytoscape software.

### 2.9. Immune Cell Infiltration Analysis

CIBERSORT is an algorithm that can estimate cellular composition in complex tissues based on normalized gene expression data [30]. In this study, the mRNA expression profile data of cerebral cortex tissue in the HIBD group and control group were extracted, and the relative proportions of 22 immune cells in each cerebral cortical tissue were assessed via CIBERSORT. Referring to the method of Li et al. [31], the LM22 feature matrix will be applied to predict the proportion of immune cells. Moreover, histograms of immune cell proportions, boxplots of immune cell expression, boxplots of immune cell difference analysis, heat map of correlation analysis between immune cells, and heat map of correlations between optimal hub FRDEGs and immune cell infiltration were plotted, respectively.

## 3. Result

### 3.1. Determination of Ferroptosis-Related DEGs

In the GSE23317 dataset, 270, 628, and 2019 DEGs (Figure 2a) were obtained based on the comparison of gene expression in the cerebral cortex of the HIBD and sham groups at 3 h, 8 h, and 24 h, respectively. A total of 2092 up-regulated DEGs and 698 down-regulated DEGs were found after removing overlapping genes across periods. Finally, a total of 366 FRGs were obtained and screened. They were further intersected with DEGs to obtain 56 FRDEGs (Figure 2b), including 27 driver genes, 20 inhibitor genes, and 22 marker genes (Table 2). The expression of FRDEGs was displayed (Figure 2c) via a cluster heat map. A total of 59 DE miRNAs were identified in the GSE184939 dataset, including 43 up-regulated and 16 down-regulated DE miRNAs (Figure 2d).

### 3.2. Enrichment Analysis of Ferroptosis-Related DEGs

A total of 56 FRDEGs were performed for enrichment analysis. According to enrichment results of biological function (Figure 3a), these DEGs were significantly enriched in biological processes, such as cell proliferation, apoptosis regulation, gluconeogenesis regulation, and iron homeostasis. Meanwhile, the analysis of cellular components revealed that these DEGs were enriched in the cytosol, macromolecular compounds, RNA polymerase II transcription factor complexes, cytoplasm, and mitochondria. Molecular function analysis showed that these DEGs were mainly related to enzyme, protein, and transcription factor bindings. Moreover, pathway enrichment results indicated that these DEGs were mainly associated with ferroptosis, autophagy, hypoxia, metabolic, and immune-inflammatory pathways (Figure 3b). The detailed enrichment results of biological functions and signal pathways are shown in the Appendix A.

### 3.3. Construction of Co-Expression Modules and Determination of Key Modules

In this study, the gene co-expression network of the mRNA dataset was constructed by utilizing the R package of WGCNA. Specifically, soft threshold β = 11 was chosen (Figure 4a), and 20 related modules were obtained (Figure 4b). The results of the module–feature relationship are shown in Figure 4c, indicating that the salmon (r = 0.89; *p* = 2 × 10^−8^), yellow (r = 0.64; *p* = 0.001), and red (r = 0.72; *p* = 2 × 10^−4^) modules were most positively related to the occurrence of HIBD. These three modules contain 213 genes in total. The hub genes (GS > 0.2, MM > 0.8) in the above three key modules were subjected to GO biological process and KEGG enrichment analysis. According to the GO biological process enrichment analysis results, these genes were mainly enriched in biological processes, such as neuronal apoptosis regulation, transcription regulation, cell proliferation regulation, and cell cycle (Figure 4d). Moreover, the results of KEGG enrichment analysis were mainly related to signaling pathways, such as inflammatory response, apoptosis, and immune response (Figure 4e). The detailed enrichment results of biological functions and signal pathways are shown in the Appendix A. In a word, these genes screened by WGCNA have complex relationships with immunity and inflammation in biological functions and signal pathways.

### 3.4. Determination and Validation of the Optimal Hub FRDEGs

The FRDEGs were subjected to PPI network analysis. A total of 28 hub DEGs (PPI-hub DEGs) were obtained according to the screening criteria (Figure 5a). All PPI-hub DEGs were intersected with WGCNA-hub genes to obtain eight overlapping DEGs (Figure 5b), which were defined as the optimal hub FRDEGs. The GSE144456 and GSE112137 datasets were used for validation, which tested the screening results’ reliability further. The expression of the optimal hub FRDEGs in different periods for mouse and human cerebral cortical cells is shown in histograms (Figure 5c–e). Furthermore, the interrogation of transcriptome data showed that optimal hub FRDEGs were also expressed in the human term brain in contrast to the preterm one, which eliminated the interference on different development stages for gene expression (Figure 5g). Based on the above results, all genes except Gpt2 were differentially expressed in the cerebral cortex samples of mice and humans at different ages. Their expression trends in GSE23317 were consistent, indicating that the screening results were relatively reliable. Therefore, these seven optimal hub FRDEGs were selected for further analysis. In addition, expression pattern correlation analysis showed that the expression of Jun, Atf4, and Stat3 strongly positively correlated with that of Ddit3, Ddit4, and Slc2a1 (Figure 5f). In general, the stability of expression and the correlation of expression patterns of these FRDEGs in HIBD models of term and preterm infants, to a certain extent, proved the close relationship between ferroptosis-related genes and HIBD.

### 3.5. miRNA Prediction

MiRNAs of the optimal hub FRDEGs were predicted via the miRWalk platform, and the following criteria were used to filter the results: *p* (accessibility) < 0.05, Binding Score > 0.95, 3′ UTR as the target gene binding region, and then overlapping miRNAs were obtained with DEmiRNAs in GSE184939 to ensure the accuracy and reliability of our results. Figure 6 shows the constructed gene-miRNA network with 28 miRNAs (22 up-regulated, six down-regulated), seven hub genes, and 60 edges. Among them, four key miRNAs (miR-7032-3p, miR-150-3p, miR-221-5p, and miR-709) were screened according to Degree ≥ 4 (Figure 6), which reveals that these miRNAs regulate more Hub FRDEGs, indicating their core regulatory position.

### 3.6. Immune Infiltration Analysis

The proportions and infiltrating abundances of the 22 detected immune cells in each sample are shown in histograms (Figure 7a) and boxplots (Figure 7b) via the CIBERSORT algorithm, with good results, and their correlation is shown in (Figure 7c). Specifically, the activation of hypertrophic cells is strongly positively correlated with M2 macrophages, Gamma-Delta T cells and naive CD4+ T cells, and M2 macrophages and naive CD4+ T cells are strongly positively correlated. Immune cell variation analysis showed (Figure 7d) that 10 cell types were significantly different in HIBD cortical tissue compared with control group. In HIBD group, the proportion of hypertrophic cells, M2 macrophages, activated NK cells, naive CD4 + T cells and gamma delta T cells increased, while the proportion of resting hypertrophic cells, M0 macrophages, resting NK cells, naive B cells, activated CD4 + T memory cells, and adjusted T cells decreased (*p* < 0.05). Gene-immune cell correlation analysis showed (Figure 7e) that the expressions of Stat3, Slc2a1, and Ddit3 were strongly positively correlated with the activation of NK cells, while the expression of Slc2a1 was strongly positively correlated with the activation of mast cells. These results further support that the expression level of seven optimal iron death related genes may affect the immune activity of immune cells.

## 4. Discussion

The pathological process of ferroptosis has been reported to share some features with HIBD, such as iron overload and elevated lipid peroxides [32,33,34]. Recent studies have shown that targeted ferroptosis therapy, such as the antioxidant carvacrol, can inhibit the occurrence of ferroptosis in hippocampal neurons by reducing lipid peroxide levels in ischemic brain tissue, thereby ameliorating the impairment of learning and memory ability caused by cerebral ischemia [35]. In addition, the ferroptosis-specific inhibitor ferrostatin-1 can inhibit the occurrence of neuronal ferroptosis by reducing the production of lipid ROS and the expression of cyclooxygenase 2 [36]. However, the mechanism of ferroptosis in HIBD has yet to be thoroughly studied. Therefore, this study sought to identify biomarkers associated with neonatal HIBD ferroptosis to explore their role in immune cell infiltration in HIBD. To the best of our knowledge, this is the first bioinformatic report describing the coexistence of ferroptosis and immune cell infiltration in the pathogenesis of neonatal HIBD.

Based on differential gene expression analysis, this study identified 56 genes that may be involved in ferroptosis in HIBD. Enrichment analysis showed an overall association between ferroptosis and neonatal HIBD. In addition, FRDEGs related to functions and pathways such as autophagy, hypoxia response, metabolic pathways, and immune inflammation were also significantly enriched in neonatal HIBD. According to the enrichment analysis of key genes obtained by WGCNA analysis, HIBD is mainly involved in the inflammatory response, apoptosis, immune response, and lipid metabolism. Overall, the results of this study demonstrate that ferroptosis is activated in neonatal HIBD, and links of neonatal HIBD to these biological processes and pathways have been reported in previous studies [37,38], suggesting that our findings are robust. A large number of in vitro experiments have revealed that iron homeostasis plays an important role in regulating immune and inflammatory responses. For example, after ferroptosis occurs, cells will release many factors that can activate innate immunity, such as DAMPs to activate inflammatory signaling pathways, release inflammatory factors, recruit inflammatory cells, and expand the inflammatory response [39]. Autophagy is also involved in the progression of neonatal HIBD. Studies have shown that autophagy regulates intracellular iron homeostasis and the production of reactive oxygen species, which play a central role in the induction of ferroptosis [40].

In this study, seven optimal ferroptosis-related hub genes (Jun, Ddit3, Ddit4, Atf4, Slc2a1, Stat3, and Slc40a1) were further screened out, and there is a strong correlation between their expression levels. Jun is a proto-oncogene. Meanwhile, studies have found that overexpression of c-JUN can promote the synthesis of glutathione, thereby inhibiting ferroptosis [41]. In addition, ferroptosis can induce endoplasmic reticulum (ER) stress. The ER stress indicator Atf4 is a leucine zipper transcription factor that can regulate multiple unfolded protein response (UPR) target genes [42]. Studies have shown that Atf4 can increase the expression of the transcriptional activator Ddit3, thereby promoting apoptosis and reducing neuronal survival after intracerebral hemorrhage [43]. Other studies indicated that Stat3 could promote the expression and release of cathepsin B, mediate lysosomal dysfunction, and promote ferroptosis [44]. Deficiency of the Slc2a1 gene and its translation product (the glucose transporter 1 protein, GLUT1) will impair glucose uptake and affect brain function, thereby leading to neurodevelopmental disorders [45]. According to previous studies, the downregulation of Slc40a1 leads to ferrugination and ferroptosis in the brain, with subsequent cognitive impairment in patients with type 1 diabetes [46]. Collectively, these seven ferroptosis-related genes are involved in the pathogenesis of brain injury in neonatal HIBD. The expression levels of seven central genes were validated in the GSE144456 and GSE112137 datasets. That is, these ferroptosis-related genes showed similar expression trends in term and preterm animal HIBD models, as well as in human cell hypoxic injury models. In addition, the miRNAs that regulated them were further predicted, and four key miRNAs (miR-7032-3p, miR-150-3p, miR-221-5p, and miR-709) were obtained via topological analysis. Previous studies have shown that miR-150-3p enhances neuroprotective effects of neural stem cell exosomes after hypoxic-ischemic brain injury by targeting CASP2 [47]. Another similar study showed that targeting miR-709 promotes inflammatory response and secondary brain injury after intracerebral hemorrhage [48]. Based on the above literature and the analysis results of this study, regulating the occurrence and progression of ferroptosis in neonatal HIBD via targeted miRNAs is promising.

Immune cells have been reported to be involved in the pathogenesis of neonatal HIBD, and circulating immune cell activation is associated with poor outcomes in brain injury [49]. In this study, immune cell infiltration analysis was performed on the cerebral cortex of the neonatal HIBD model, and the results showed that there might be a certain regulatory relationship between different immune cells. In addition, this study also revealed the dysregulation of 10 immune cells, including hypertrophic cells, M2 macrophages, activated NK cells, naive CD4+ T cells, and Gamma-Delta T cells in the cerebral cortex of the HIBD group. The association between ferroptosis and immune infiltration is extremely complex. It has been confirmed that lymphocytes have a protective role in neonatal encephalopathies, such as decreased T cells increase inflammatory cell infiltration, primarily neutrophils, and inflammatory macrophages, which contribute to increased HI-induced brain damage, indicating that regulatory T cells in the injured brain may be an important mechanism of endogenous neuroprotection [50,51]. Previous studies have shown that the cytotoxicity in neonatal NK cells and low expression levels of L-selectin and CD54 lead to impaired ability to adhere to target cells [52]. In addition, the inactivation of NK cells can reduce brain and systemic organ atrophy and neurobehavioral deficits [53]. Significantly, some studies have found that the proportion of inflammatory cells in the blood can be used as an evaluation index for the risk of HIBD and the efficacy of mild hypothermia [54]. Significantly, ferroptosis-related genes Stat3, Slc2a1, Ddit3, and Slc40a1 are involved in regulating the HIBD micro immune environment. Currently, from the above findings, there is a significant correlation between ferroptosis and immune cell infiltration. It indirectly means that ferroptosis may promote the occurrence and development of neonatal HIBD by activating immune infiltration and responses.

There still exist some limitations in this research. First, further vivo experiments, to confirm a more detailed association between ferroptosis related genes and histopathology, regional distribution, and upstream regulators in neonatal HIBD were lacking to validate the results. Secondly, the exact mechanism of ferroptosis-related genes and immune cell infiltration requires further study. The last one is that CIBERSORT analysis is based on limited genetic data that may deviate from heterotypic interactions of cells, disease-induced disease, or phenotypic plasticity. Similarly, these research results should be used with caution. Nonetheless, this study provides a basis for further research. Most remarkably, the occurrence and development of HIBD are complex and multifactorial, which is an essential obstacle to pathogenesis research. In future studies, an in-depth analysis of the above specific regulatory relationship and mechanisms in HIBD based on time of onset and clinical prognosis should be performed.

## 5. Conclusions

In this study, bioinformatics analysis was used to identify hub FRDEGs that are closely related to the development of HIBD (Jun, Ddit3, Ddit4, Atf4, Slc2a1, Stat3, and Slc40a1) after 56 FRDEGs in neonatal HIBD were screened out. In addition, the infiltration of immune cells of the cerebral cortex in HIBD and the interaction between immune cells were also revealed. In particular, it was discovered that some ferroptosis-related genes exhibited high positive correlations with different cell types. Since there are no mature and reliable therapeutic drugs for neonatal HIBD and combined with our research results, targeted regulation of ferroptosis and immune cell infiltration has considerable prospects for treatment.

## Figures and Tables

**Figure 1 cells-11-03778-f001:**
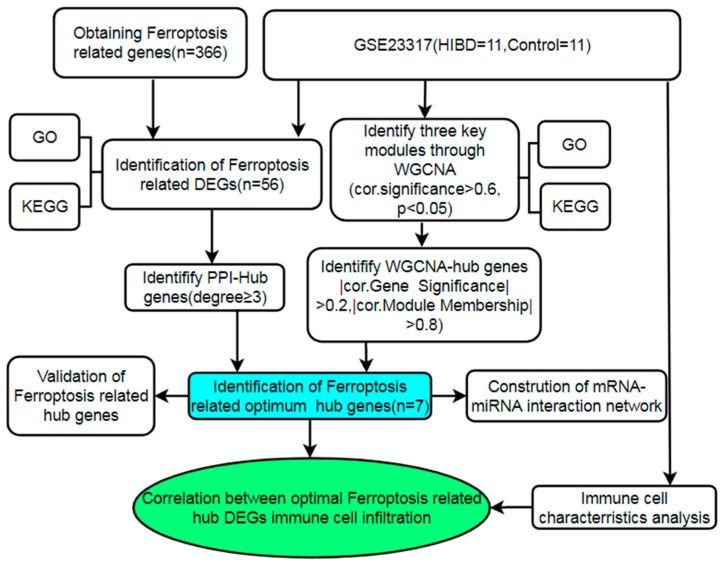
Flow chart of investigations in this study.

**Figure 2 cells-11-03778-f002:**
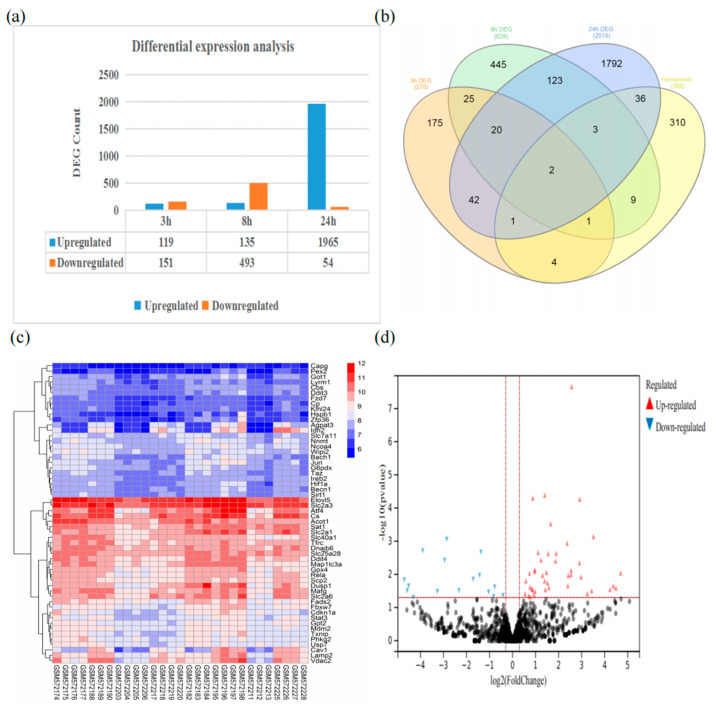
(**a**) Results of mRNA differential expression analysis; (**b**) Venn diagram of the relationship between DEGs at different times and ferroptosis-related genes; (**c**) Heat diagram of FRDEGs; (**d**) MiRNA differential expression analysis (black and gray dots represent miRNA with no significant differential expression).

**Figure 3 cells-11-03778-f003:**
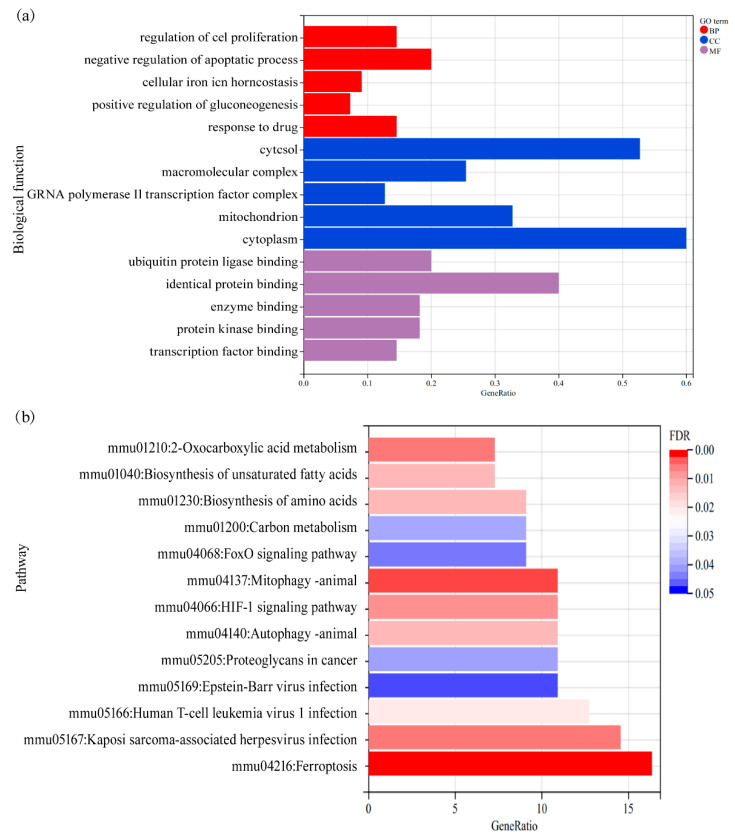
Results of molecular function and pathway enrichment analysis. (**a**) Function enrichment analysis (GO: gene ontology, BP: biological process, CC: cell component and MF: molecular function); (**b**) signal pathway enrichment analysis. FDR: false discovery rate.

**Figure 4 cells-11-03778-f004:**
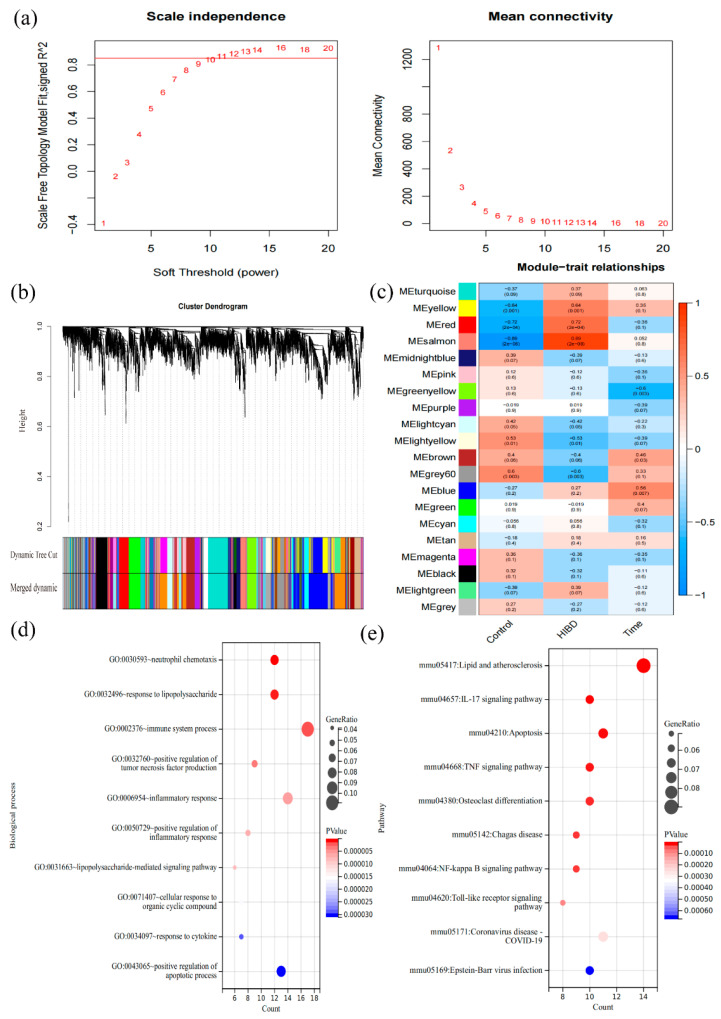
The co-expression modules analysis. (**a**) The relationship between the scale-free fit index and various soft-thresholding powers; (**b**) Clustering dendrogram of genes, various colors represent different modules; (**c**) The relationship of 3 traits and 20 modules; (**d**) Results of enrichment analysis of biological processes; (**e**) Results of enrichment analysis of signal pathways.

**Figure 5 cells-11-03778-f005:**
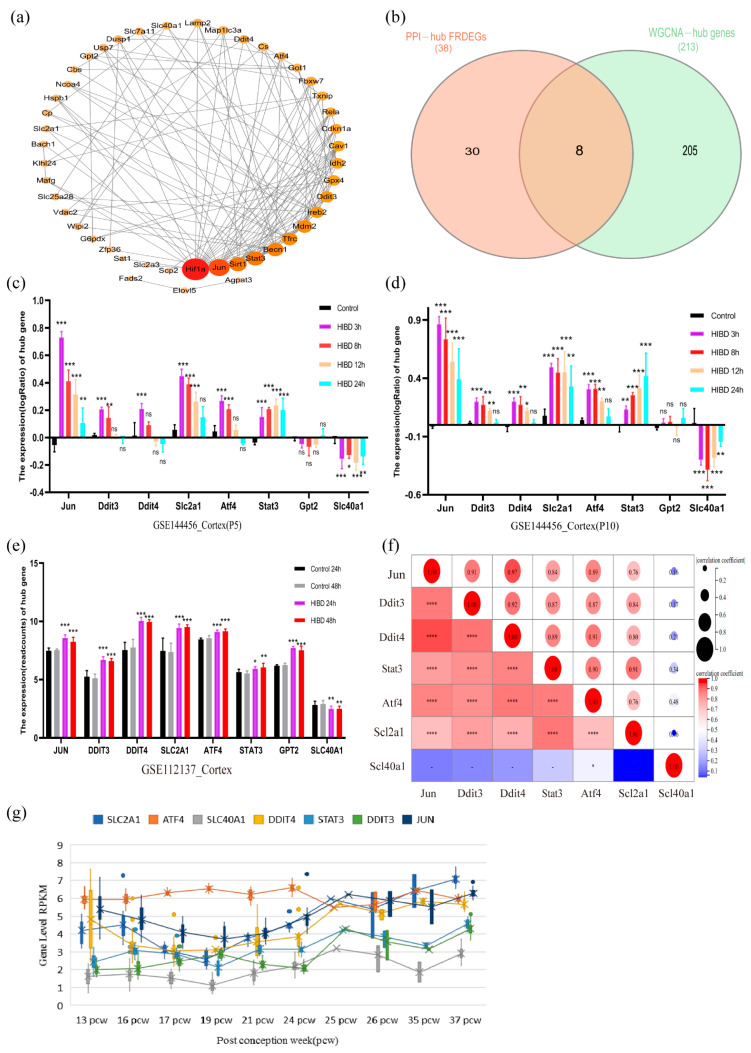
(**a**) PPI network of 56 FRDEGs; (**b**) The screening of the optimal ferroptosis-related genes; (**c**,**d**) The expression levels of the optimal optimal hub FRDEGs in different ages mouse models; (**e**) The expression levels of the optimal optimal hub FRDEGs in the human cerebral cortex models; (**f**) Correlation analysis of expression patterns of the optimal hub FRDEGs. (**g**) Dynamic expression of the optimal hub FRDEGs in brain tissue development from 13 weeks of gestation (preterm) to 37 weeks of gestational age (full-term). Note: ns (*p* > 0.05), * (*p* ≤ 0.05), ** (*p* ≤ 0.01), *** (*p* ≤ 0.001), **** (*p* ≤ 0.0001).

**Figure 6 cells-11-03778-f006:**
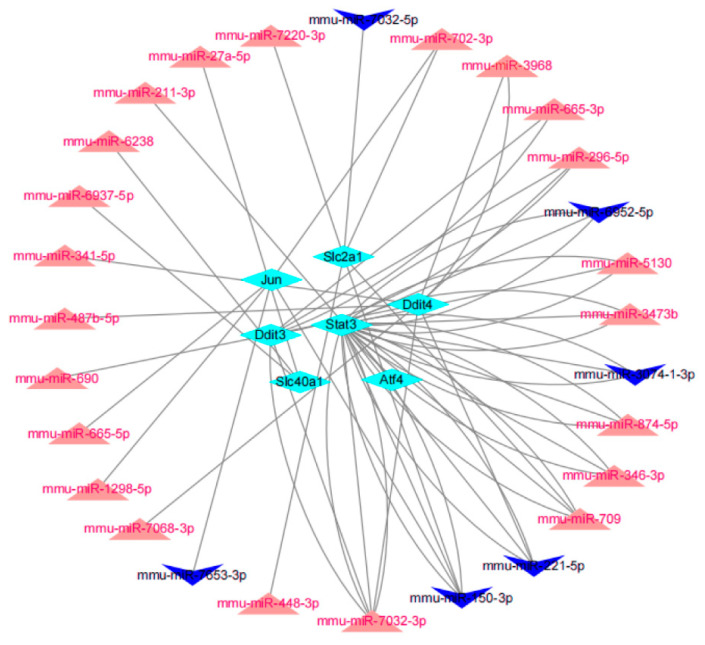
The regulatory network of the optimal hub FRDEGs-miRNA. Red represents up-regulated miRNA, blue represents down-regulated miRNA, and turquoise represents the optimal hub FRDEGs.

**Figure 7 cells-11-03778-f007:**
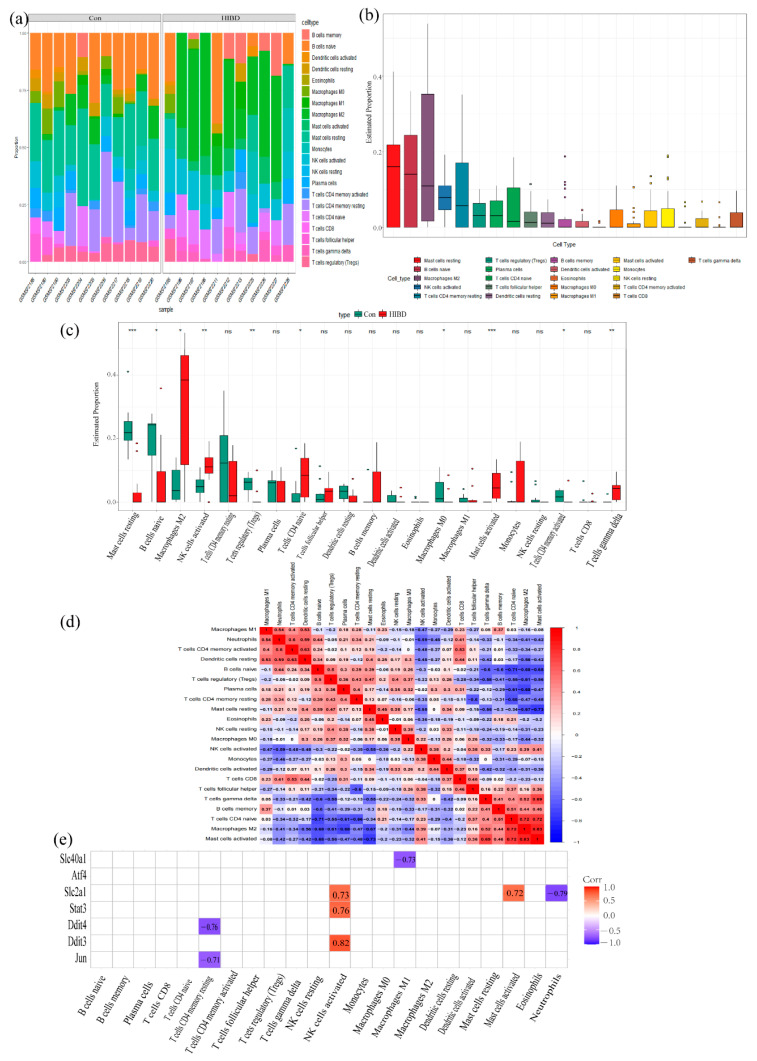
Immune cell infiltration analysis. (**a**) The proportions of the 22 detected immune cells in each sample; (**b**) infiltrating abundances of immune cell infiltration; (**c**) Difference of immune cells in cerebral cortex between HIBD model and control group; (**d**) Correlation analysis between infiltrated immune cells; (**e**) Relationship between the optimal hub FRDEGs and immune infiltration. Note: ns (*p* > 0.05), * (*p* ≤ 0.05), ** (*p* ≤ 0.01), *** (*p* ≤ 0.001).

**Table 1 cells-11-03778-t001:** The detailed characteristics of the datasets.

Dataset	Platform	Species	No. of Samples (Control/HIBD)	Type	Tissue	Experimental Model
GSE23317	GPL6885	Mus musculus	P8: 3 h: 4/3 8 h: 3/4 24 h: 4/4	mRNA	Cerebral cortex	Unilateral carotid artery ligation+hypoxia (8% oxygen/92% N_2_, 1 h)
GSE144456	GPL10333	Mus musculus	P5, P10: 3 h: 3/3 6 h: 3/3 12 h: 3/3 24 h: 3/3	mRNA	Forebrain	Unilateral carotid artery ligation+hypoxia (8% oxygen, 40 min)
GSE112137	GPL20301	Homo sapiens (extreme prematurity)	24 h: 8/8 48 h: 8/8	mRNA	Cortical progenitors	At day 74–78 of in vitro differentiation, hCS are exposed for 24 h and 48 h at <1% O_2_ in a gas-controlled culture chamber. Control hCS are maintained at 21% O_2_ throughout.
GSE184939	GPL21626	Mus musculus	P9: 4/4	microRNA	Cerebral cortex	Unilateral carotid artery ligation+hypoxia (8% oxygen, 30 min)

**Table 2 cells-11-03778-t002:** Details of DEGs related to Ferroptosis.

Type	DEGs
Driver	Ireb2, Cs, Vdac2, Scp2, Tfrc, Go1t, Ncoa4, Phkg2, G6pdx, Becn1, Map1lc3a, Wipi2, Sat1, Hif1a, Taz, Sirt1, Fbxw7, Dnajb6, Bach1, Elovl5, Usp7, Aat4, Apat3, Pex2, Lyrm1, Slc25a28,Mdm2
Suppressor	Slc7a11, Gpx4, Hspb1, Slc40a1, Atf4, Fads2, Stat3, Cdkn1a, Vdac2, Cbs, Hif1a, Jun, Lamp2, Zfp36, Cav1, Fzd7, Acot1, Cp, Idh2, Rela
Marker	Dusp1, Slc7a11, Ddit4, Ddit3, Txnip, Gpt2, Cbs, Atf4, Klhl24, Rela, Agpat3, Tfrc Mafg, Slc40a1, Gpx4, Hspb1, Slc2a1, Slc2a3, Slc2a6, Ireb2, Nnmt, Capg

Driver: a gene that promotes ferroptosis; Suppressor: a gene that prevents ferroptosis; Marker: a gene that indicates ferroptosis occurrence.

## Data Availability

The data used to support the findings of this study have been uploaded to the Appendix A. More detailed data can be obtained from the corresponding author.

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
