# Peer review of "New Insights into Mechanisms of Ferroptosis Associated with Immune Infiltration in Neonatal Hypoxic-Ischemic Brain Damage"

_cells, 2022, doi:10.3390/cells11233778_

Round 1

Reviewer 1 Report

Li and colleagues contribute a manuscript on the supervised analysis of microarray datasets for ferroptosis related genes in a neonatal mouse model of hypoxic-ischemic brain damage. They used datasets collected from the GEO database. Four datasets were analyzed (3 mRNA datasets and 1 miRNA). Gene ontology and Kyoto encyclopedia of gene pathway enrichment analyses were done. Genes related to immune cell infiltration were also analyzed. They report that 56 differentially expressed  genes are ferroptosis-related. Immune cell infiltration genes and ferroptosis differentially expressed genes were positively correlated.

A strength of this manuscript is the demonstration of the use of existing datasets of differential expressed genes from experimental models and the interpretations derived from such analyses.

This manuscript has important weaknesses.

11.       The development of the rationale for this study is not strong. The authors interrogate microarray datasets for the differential expression of ferroptosis related genes. The existence and robustness of ferroptosis in neonatal brain injury is not evident. Gou et al 2020 is cited in this regard, but the work is not very convincing. Moreover, Gou et al use a neonatal rat model and is on hippocampus. This manuscript by Li et al uses neonatal mouse and focuses on hippocampus. There are important mismatches in the logical design. The authors need to build a case for how common ferroptosis is in neonatal mouse hypoxic-ischemic brain damage, where it occurs, and when it occurs.

22.   The details of some of the datasets are wanting in explanation. For example, the human microarray dataset (Table 1) needs to be explained better.  

33. The experimental design is not properly controlled. Because the injury model is a unilateral vessel occlusion, the important comparison is ipsilateral cerebral cortex versus contralateral cerebral cortex. They need to control for the effects of hypoxia.

44. The authors provide no association between the differentially expressed genes and the neuropathological/structural evidence for ferroptosis.

55. There is no attempt to verify the differential expression of mRNA by RT-PCR or protein level analyses.

66.  They could have confirmed that some of the ferroptosis related mRNAs were in microglia cells or vascular derived immune cells.

77. Many of the differentially expressed genes that are classified as ferroptosis related (Table 2) encode multifunctional proteins. So the categorization of the genes is subjective and biased.

   8. The cerebral cortex is not uniformly affected in this animal model. It is not clear where the changes in differentially expressed genes are occurring.      

Author Response

Responses to the Reviewer1’ Comments

We would like to thank you for the constructive comments. We have made our best effort to address all the issues raised your comments, and revised the manuscript thoroughly and carefully. The point-by-point responses to the comments are provided below, and all the major changes are highlighted in the revised manuscript. All your suggestions are very constructive and professional. However, we have difficulties in further verification: firstly, the city where our team works is experiencing a large-scale epidemic of novel coronavirus, and experimental equipment and reagents cannot be obtained smoothly. Recently, all our members have been engaged in the task of combating the COVID-19 epidemic. There are not enough people at our disposal. Secondly, We are applying for a project on the relationship between iron death and HIBD, and have not yet received a reply. In addition, there are considerable difficulties in supplementing relevant verification when animal ethical issues are involved.

Therefore, we sincerely agree with your comments, but we look forward to more feasible suggestions for improving the quality of our articles. Our results have been verified in two datasets, improving the reliability to a certain extent. We are also trying our best to replenish the corresponding validation experiments, which may take a considerable amount of time to wait.

11.The development of the rationale for this study is not strong. The authors interrogate microarray datasets for the differential expression of ferroptosis related genes. The existence and robustness of ferroptosis in neonatal brain injury is not evident. Gou et al 2020 is cited in this regard, but the work is not very convincing. Moreover, Gou et al use a neonatal rat model and is on hippocampus. This manuscript by Li et al uses neonatal mouse and focuses on hippocampus. There are important mismatches in the logical design. The authors need to build a case for how common ferroptosis is in neonatal mouse hypoxic-ischemic brain damage, where it occurs, and when it occurs.

Response: Thank you for your professional issue. We reviewed the literature and added new citations discussing the current ferroptosis correlation with HIBD in newborns. As far as we know, existing studies have shown that ferroptosis-related mechanisms are found in the cerebral cortex and hippocampus of HIBD. Studies have found that, after hypoxia-ischemia (HI), the dynamic changes of iron distribution, iron content, and malondialdehyde (MDA) in different brain regions (parietal cortex, corpus callosum, hippocampus) of neonatal rats within 84 days may cause lipid peroxidation to damage brain tissue. This property suggests that there is a relationship between abnormal iron metabolism and tissue damage in HIBD[1]. Recent research shows abnormal regulation of iron metabolism, amino acid metabolism, and lipid peroxidation in HIBD leads to the decreased antioxidant capacity of neurons and mitochondrial damage, leading to iron death of neurons in the cerebral cortex[2]. Another study also showed that the activation of TLR4 induces Ferroptosis of HIBD neurons; on the contrary, the inhibition of TLR4 improves neuroinflammation[3]. Moreover,studies have confirmed that neuronal ferroptosis was observed in the neonatal HIBD rat model, and the injection of ferroptosis inhibitor melatonin can significantly inhibit neuronal ferroptosis, promote the survival of hippocampal neurons, and improve learning and memory abilities in HIBD rats[4]. In conclusion, previous studies have shown that abnormal iron metabolism and ferroptosis are involved in the pathogenesis of HIBD. More studies are required to explore the presence of ferroptosis in more sites gradually.

Our previous description was not clear enough. Most of the datasets we included were from cerebral cortex samples, focusing on the relationship between ferroptosis in the cerebral cortex and HIBD. No dataset for studying HIBD hippocampus has been published in GEO database.

[1] Hu DW, Zhang G, Lin L, Yu XJ, Wang F, Lin Q. Dynamic Changes in Brain Iron Metabolism in Neonatal Rats after Hypoxia-Ischemia. J Stroke Cerebrovasc Dis. 2022 Apr;31(4):106352.

[2] Lin W, Zhang T, Zheng J, Zhou Y, Lin Z, Fu X. Ferroptosis is Involved in Hypoxic-ischemic Brain Damage in Neonatal Rats. Neuroscience. 2022 Apr 1;487:131-142.

[3] Zhu K, Zhu X, Sun S, Yang W, Liu S, Tang Z, Zhang R, Li J, Shen T, Hei M. Inhibition of TLR4 prevents hippocampal hypoxic-ischemic injury by regulating ferroptosis in neonatal rats. Exp Neurol. 2021 Nov;345:113828.

[4] Gou Z, Su X, Hu X, Zhou Y, Huang L, Fan Y, Li J, Lu L. Melatonin improves hypoxic-ischemic brain damage through the Akt/Nrf2/Gpx4 signaling pathway. Brain Res Bull. 2020 Oct;163:40-48.

22.The details of some of the datasets are wanting in explanation. For example, the human microarray dataset (Table 1) needs to be explained better.

Response: Thank you for your suggestion of modification. According to the research of PaÅŸca et al.[5], GSE112137 contains gene expression data after 24 hours and 48 hours of hypoxia treatment on human cortical spheroids (hCS, differentiated from human induced pluripotent stem cells) that transcriptionally resembled the cerebral cortex at 19–24 PCW.

[5]Pașca AM, Park JY, Shin HW, et al. Human 3D cellular model of hypoxic brain injury of prematurity. Nat Med. 2019 May;25(5):784-791.

The table after supplementary information is as follows

Table 1. The detailed characteristics of the datasets.

Dataset

Platform

Species

No. of samples (Control/HIBD)

Type

Tissue

Experimental model

GSE23317

GPL6885

Mus musculus

P8:

3h: 4/3

8h: 3/4

24h: 4/4

mRNA

Cerebral

contex

Unilateral carotid artery ligation+hypoxia(8% oxygen/92%N 2, 1 hour)

GSE144456

GPL10333

Mus musculus

P5,P10:

3h: 3 /3

6h: 3/3

12h: 3/3

24h: 3/3

mRNA

Forebrain

Unilateral carotid artery ligation+hypoxia(8% oxygen, 40minutes)

GSE112137

GPL20301

Homo sapiens (extreme prematurity)

  24h: 8/8

  48h: 8 /8

mRNA

Cortical

progenitors

At day 74–78 of in vitro differentiation, hCS are exposed for 24hours and 48 hours at <1% O2 in a gas-controlled culture chamber. Control hCS are maintained at 21% O2 throughout.

GSE184939

GPL21626

Mus musculus

P9: 4/4

microRNA

Cerebral

contex

Unilateral carotid artery ligation+hypoxia(8% oxygen, 30minutes)

  1. The experimental design is not properly controlled. Because the injury model is a unilateral vessel occlusion, the important comparison is ipsilateral cerebral cortex versus contralateral cerebral cortex. They need to control for the effects of hypoxia.

Response: Thank you for raising this controversial issue. Notably, unlike ischemic brain injury in adults, neonatal hypoxic-ischemic encephalopathy is a kind of tissue hypoxia-ischemia caused by asphyxia, not tissue hypoxia-ischemia caused by ischemia. The most commonly used neonatal HIBD experimental model is the method of unilateral typical carotid artery occlusion followed by systemic hypoxia in neonatal rats[6]. Most of the data sets we selected were neonatal HIBD models made in this way. Although there are differences in the specific treatment of models, the stable expression of these screened genes in these HIBD models also shows the reliability of the results and discovery to some extent. That is, ferroptosis-related genes and their mechanisms exist in any HIBD model.

  • Rice JE 3rd, Vannucci RC, Brierley JB. The influence of immaturity on hypoxic-ischemic brain damage in the rat. Ann Neurol. 1981 Feb;9(2):131-41. doi: 10.1002/ana.410090206.

44.The authors provide no association between the differentially expressed genes and the neuropathological/structural evidence for ferroptosis.

Response: The question put forward by the reviewers is very professional, and we have also considered it. Due to the limitations of the original design, we were unable to conduct an in-depth correlation analysis. We are applying for the subject of ferroptosis and neonatal hypoxic-ischemic brain damage, and then we will conduct relevant experiments to confirm their association with neuropathological/structural evidence.

55.There is no attempt to verify the differential expression of mRNA by RT-PCR or protein level analyses.

Response: Thank you for your suggestions. We have used two transcriptome gene datasets from human neural cells and animal brain tissues to verify these ferroptosis-related genes from the perspective of premature and term infants. In addition, in the revised manuscript, because we only found the data on hypoxic brain damage in premature infants, we proved that these genes were expressed in full-term human infants as in premature infants, eliminating the interference of development. We are applying for a research project on ferroptosis and neonatal hypoxic-ischemic brain damage. On the premise of abiding by animal ethics, we are completing more verifications but not limited to ferroptosis-related genes.

66.They could have confirmed that some of the ferroptosis related mRNAs were in microglia cells or vascular derived

Response: Thank you for your comments, but we do not particularly understand the meaning expressed. We answer this question according to our understanding. Due to the defects of the original research, it is impossible to determine the detailed correlation and contact mechanism between ferroptosis-related genes and glial cells or peripheral immune cells. Based on bioinformatics technology, our research preliminarily explored ferroptosis and immune cell infiltration in HIBD. Your comments will guide our further research in immune cells.

  1. Many of the differentially expressed genes that are classified as ferroptosis related (Table 2) encode multifunctional proteins. So the categorization of the genes is subjective and biased.

Response: Thank you for your comments. We do not believe that this classification is subjective and biased. Maybe you don't know much about this classification. FerrDb is the world's first manually curated database for ferroptosis regulators and ferroptosis-disease associations from published journal articles[7]. This kind of evidence is generally represented by an author statement of the role of the regulator in an original article[7]. At present, the FerrDb database has been cited for 230 times, which is highly authoritative.

The database divides 259 ferroptosis-related genes into three categories based on confidence levels  and author statement : 108 drivers, 69 suppressors and 111 markers[7].

Driver: A gene that promotes ferroptosis.

Suppressor: A gene that prevents ferroptosis.

Marker: A gene that indicates ferroptosis occurrence

If there are any other questions, we are happy to answer them.

[7 ]Zhou N, Bao J. FerrDb: a manually curated resource for regulators and markers of ferroptosis and ferroptosis-disease associations. Database (Oxford). 2020 Jan 1;2020:baaa021.

8.The cerebral cortex is not uniformly affected in this animal model. It is not clear where the changes in differentially expressed genes are occurring.   

Response: Thank you for your comments. This problem is worth studying. Most of the data samples included in this study were mainly from the cerebral cortex of the HIBD model, and needed to be more accurate to the specific location of the cortex. This is also the limitation of this study. Our team accepted the reviewers' suggestions and will conduct in-depth research on the subject being applied. In general, our study revealed the macro correlation between ferroptosis-related genes and neonatal HIBD. We pointed out in the discussion that this proposal is very constructive for us and even for other researchers in this field in the future.

Reviewer 2 Report

This study has used several gene expression data bases and bioinformatics to build a picture regarding molecular pathways and interactions associated with ferroptosis and the immune response following hypoxia-ischaemia in mice and human neonates.  As expected, this study has generated a plethora of data and interactions with different genes/proteins, which while complicated does provide useful information regarding pathogenic processes associated with hypoxic-ischeamic brain injury.  In addition, while the data can be quite overwhelming scientists that have an interest in the molecular mechanisms and cellular pathways involving hypoxic-ischaemic brain injury or interest particular molecules identified will find the study useful and informative. Therefore, the study is worthy of publication.

I have provided some suggestions to help improve the manuscript.

Introduction

This sentence needs context and improving: “In addition, the proportion imbalance of immune cells is related to brain damage and function loss in premature infants after hypoxia and ischemia[11].

Also are the authors specifically referring to premature infants here. 

Provide full name for “PAI-1”.

Material and Methods

Should more information be provided regarding the hypoxic-ischaemia model used to generate the gene/miRNA data sets. 

Please refer to Table 1 in Material and Methods for details of data sets.

Consider moving Section “3.1 Dataset character” (change to “Dataset characteristics”) to Material and Methods Section, as this section does not contain results generated by the study.

Table 1: What was age of mice for GSE23317 and GSE184939 data sets.  Better define ?? vs ?? in table.  Correct “contex”.

Results

Table 2; I am unfamiliar with terms “driver”, “inhibitor”, and “marker” genes. Could these be defined??

Table 2: improve spacing after commas.

Omit “etc”.

Figure 3a: Define GO term and BP, CC, MF in figure legend.

Figure 3b: Define FDR in figure legend.

Figure 7: Size/quality of labels on some figures needs to be enlarged/improved.

Discussion

Define “UPR”

Author Response

Responses to the Reviewer2’ Comments

We would like to thank you for the constructive comments. We have made our best effort to address all the issues raised your comments, and revised the manuscript thoroughly and carefully. The point-by-point responses to the comments are provided below, and all the major changes are highlighted in the revised manuscript. All your suggestions are very constructive and professional.

 Introduction

 This sentence needs context and improving: “In addition, the proportion imbalance of immune cells is related to brain damage and function loss in premature infants after hypoxia and ischemia[11].

Also are the authors specifically referring to premature infants here. 

Response: Thank you for raising this question. This is because our explanation is not comprehensive enough. First of all, newborns include term infants and premature infants and Hypoxic-ischemic encephalopathy (HIE) is one of the most important causes of brain injury in preterm infants[1]. Preterm HIE is predominantly caused by global hypoxia-ischemia . There is no doubt that hypoxic-ischemic brain damage in neonates studied in this paper should include premature infants.

In addition, we have modified this description to avoid similar problems.

[1]Ristovska S, Stomnaroska O, Danilovski D. Hypoxic Ischemic Encephalopathy (HIE) in Term and Preterm Infants. Pril (Makedon Akad Nauk Umet Odd Med Nauki). 2022 Apr 22;43(1):77-84.

Provide full name for “PAI-1”.

Response: Full name for “PAI-1” is “Plasminogen activator inhibitor-1”

Material and Methods

Should more information be provided regarding the hypoxic-ischaemia model used to generate the gene/miRNA data sets. 

Please refer to Table 1 in Material and Methods for details of data sets.

Response: We are sorry for our incomplete description. We checked the information of the data set for appropriate supplement.

The table1 after supplementary information is as follows

Table 1. The detailed characteristics of the datasets.

Dataset

Platform

Species

No. of samples (Control/HIBD)

Type

Tissue

Experimental model

GSE23317

GPL6885

Mus musculus

P8:

3h: 4/3

8h: 3/4

24h: 4/4

mRNA

Cerebral

contex

Unilateral carotid artery ligation+hypoxia(8% oxygen/92%N 2, 1 hour)

GSE144456

GPL10333

Mus musculus

P5,P10:

3h: 3 /3

6h: 3/3

12h: 3/3

24h: 3/3

mRNA

Forebrain

Unilateral carotid artery ligation+hypoxia(8% oxygen, 40minutes)

GSE112137

GPL20301

Homo sapiens (extreme prematurity)

  24h: 8/8

  48h: 8 /8

mRNA

Cortical

progenitors

At day 74–78 of in vitro differentiation, hCS are exposed for 24hours and 48 hours at <1% O2 in a gas-controlled culture chamber. Control hCS are maintained at 21% O2 throughout.

GSE184939

GPL21626

Mus musculus

P9: 4/4

microRNA

Cerebral

contex

Unilateral carotid artery ligation+hypoxia(8% oxygen, 30minutes)

Consider moving Section “3.1 Dataset character” (change to “Dataset characteristics”) to Material and Methods Section, as this section does not contain results generated by the study.

Response: Thank you for your valuable comments. We have revised and supplemented according to the suggestions.

Table 1: What was age of mice for GSE23317 and GSE184939 data sets.  Better define ?? vs ?? in table.  Correct “contex”.

Response:Thank you for your comments. mice aged 8 days (P8) were obtained from GSE23317; and mice aged 9 days (P9) were obtained from GSE23317. We changed the description of this ?? vs ?? in table, Also change “contex” to “cortex”. See Table 1 for details.

Results.

Table 2; I am unfamiliar with terms “driver”, “inhibitor”, and “marker” genes. Could these be defined??

Response:Thank you for your question. FerrDb is the world's first manually curated database for ferroptosis regulators and ferroptosis-disease associations from published journal articles[2]. This kind of evidence is generally represented by an author statement of the role of the regulator in an original article[2]. At present, FerrDb database has been cited for 230 times, which is highly authoritative.

The database divides 259 ferroptosis-related genes into three categories based on confidence levels  and author statement : 108 drivers, 69 suppressors and 111 markers[2].

Driver: A gene that promotes ferroptosis.

Suppressor: A gene that prevents ferroptosis.

Marker: A gene that indicates ferroptosis occurrence

[2 ]Zhou N, Bao J. FerrDb: a manually curated resource for regulators and markers of ferroptosis and ferroptosis-disease associations. Database (Oxford). 2020 Jan 1;2020:baaa021.

Table 2: improve spacing after commas.

Response: Thank you for correcting the format error. We have corrected the spacing in Table 2. See Table 2 for details

Omit “etc”.

Response: Thank you for your suggestion. We have deleted the "etc" in the revised article.

Figure 3a: Define GO term and BP, CC, MF in figure legend.

Response: Thank you for your proposed changes.

Go term has already been defined in methodology, so it is not described in the figure legend.

Gene Ontology (GO) term is a standard glossary term for biological function annotation, which divides gene function into three parts:

Biological Process(BP) : Biological functions associated with gene products;

Molecular Function(MF): Tasks performed by gene products;

Cellular component(CC) : Location for recording the production activity of gene products[3].

We have briefly defined it in the revised article.

  • Ashburner M, Ball CA, Blake JA, et al. Gene ontology: tool for the unification of biology. The Gene Ontology Consortium. Nat Genet. 2000 May;25(1):25-9.

Figure 3b: Define FDR in figure legend.

Response: Thank you for your proposed changes. FDR, false discovery rate, obtained by using the Benjamin-Hochberg method[4] to correct the difference significance p value.

We have briefly defined it in figure legend.

  • Hochberg Y, Benjamini Y. More powerful procedures for multiple significance testing. Stat Med. 1990; 9: 811–118.

Figure 7: Size/quality of labels on some figures needs to be enlarged/improved.

Response: Thank you for your proposed changes. We have improved the size/quality of the label on Figure 7. See Figure 7 in the article for details.

Reviewer 3 Report

The involvement of ferroptosis in the progression of injury following birth asphyxia has only recently emerged. Therefore studies investigating this pathway are timely and are much needed in terms of development of therapeutics. Here Li and colleagues mine the GEO database and identify relevant microarray data to determine whether expression of ferroptosis-related genes is altered following injury. The work presented here is novel, however the authors might consider the following to further strengthen the manuscript:

General: The authors should consider providing explanatory sentences related more to the biology of their findings at the end of each section as currently the interaction of their major findings is difficult to understand. There are obvious interesting hits but for those who come at this paper from a brain injury background rather than a bioinformatics one, phrases like "Among them, 4 key miR-NAs (miR-7032-3p, miR-150-3p, miR-221-5p and miR-709) were screened according to Degree≥4" does not convey why these 4 are "key" or what that means for the biological condition. 

General: It is not obvious why the authors move from analysing ferroptosis to analysing immune cell infiltration. Can you strengthen your explanation around this section? Equally, the statement that "little is known about the mechanisms of immune cell infiltration in HIBD" is slightly misleading as there are a number of groups publishing on this (e.g. Herz et al, doi 10.1038/s41390-021-01818-7)

General: The human data set used was generated from iPScs cultured for approximately 10 weeks and representing a very immature human brain. One suggestion for an additional analysis would be to determine whether the 7 hub genes are expressed in human term brain compared with preterm brain via interrogation of the BrainSpan Atlas (brainspan.org) at the Allan Institute. This might further define whether these DEGs are more relevant to preterm or term human injury.

Introduction: The introduction would benefit from revising, to give a more detailed background of the molecular mechanism of ferroptosis as well as what is known about ferroptosis and neonatal HI. Some key references such as the first reporting of ferroptosis by Dixon et al (doi: 10.1016/j.cell.2012.03.042) or some of the more recent papers on neonatal HI and ferroptosis (e.g. Lin et al, 2022 doi: 10.1016/j.neuroscience.2022.02.013, Zui et al 2021 doi:10.1016/j.expneurol.2021.113828) would be useful.

Results: How are driver, inhibitor/suppressor and marker genes defined?

Results: Stats have been performed on the graphed data in figure 5 but a description of these tests is missing from the methods or the figure legend.

Figures: the labelling on some of the figures is too small to be legible.

Author Response

Responses to the Reviewer3’ Comments

We would like to thank you for the constructive comments. We have made our best effort to address all the issues raised your comments, and revised the manuscript thoroughly and carefully. The point-by-point responses to the comments are provided below, and all the major changes are highlighted in the revised manuscript. All your suggestions are very constructive and professional. If you have other suggestions, please help us point out.

General: The authors should consider providing explanatory sentences related more to the biology of their findings at the end of each section as currently the interaction of their major findings is difficult to understand. There are obvious interesting hits but for those who come at this paper from a brain injury background rather than a bioinformatics one, phrases like "Among them, 4 key miR-NAs (miR-7032-3p, miR-150-3p, miR-221-5p and miR-709) were screened according to Degree≥4" does not convey why these 4 are "key" or what that means for the biological condition. 

Respone: Thank you for your comment. Degree is a filter parameter of cityscape software. Degree refers to how many nodes this node is connected to. The more connected nodes, the more central this gene is in the regulatory network, indicating that it is a key gene. Nowadays, more and more attention is paid to the study of regulatory networks, because gene expression often affects the whole body, but there are always some genes that are Key genes. In Figure 6, the maximum Degree value is 6, and the minimum Degree value is 1. We select key miRNAs with the threshold of Degree>3 (≥ 4). We believe that these key miRNAs regulate a large number of ferroptosis related genes, which may play a role in overall regulation.

General: It is not obvious why the authors move from analysing ferroptosis to analysing immune cell infiltration. Can you strengthen your explanation around this section? Equally, the statement that "little is known about the mechanisms of immune cell infiltration in HIBD" is slightly misleading as there are a number of groups publishing on this (e.g. Herz et al, doi 10.1038/s41390-021-01818-7)

Response: We are sorry for our inappropriate description. Thank you for your suggestions for modification. Your suggestions will greatly help to improve the quality of research. We have modified the description of this part according to the suggestions. See page 2 for details.

Explain why the authors move from analysing ferroptosis to analysing immune cell infiltration(This part has been supplemented in introduction):

Response: Thank you for your comment. The infiltration of peripheral immune cells can affect the tissue damage, protection, repair and regeneration of HIBD. More importantly, the monitoring of brain immunity and inflammation can guide clinical decision-making. Ferroptosis is crucial in regulating the function of immune cells, mainly including that the number and function of immune cells affected by ferroptosis of immune cells and the immune inflammatory response initiated by ferroptosis of non immune cells. However, the role and signaling pathway of ferroptosis in the pathogenesis of HIBD remains elusive. 

General: The human data set used was generated from iPScs cultured for approximately 10 weeks and representing a very immature human brain. One suggestion for an additional analysis would be to determine whether the 7 hub genes are expressed in human term brain compared with preterm brain via interrogation of the BrainSpan Atlas (brainspan.org) at the Allan Institute. This might further define whether these DEGs are more relevant to preterm or term human injury.

Response: Thanks to you for your valuable suggestions. We were troubled by this problem before. We found developmental transcriptome information in the brain atlas data and confirmed the expression of seven hub genes in the brain expression profiles of both term and preterm infants. The suggestions will help us address an important flaw in the study. The results from the Brain atlas database have been supplemented in the revised manuscript(Figure 5).

Introduction: The introduction would benefit from revising, to give a more detailed background of the molecular mechanism of ferroptosis as well as what is known about ferroptosis and neonatal HI. Some key references such as the first reporting of ferroptosis by Dixon et al (doi: 10.1016/j.cell.2012.03.042) or some of the more recent papers on neonatal HI and ferroptosis (e.g. Lin et al, 2022 doi: 10.1016/j.neuroscience.2022.02.013, Zui et al 2021 doi:10.1016/j.expneurol.2021.113828) would be useful.

Response: Thanks to you for your professional suggestions and valuable literature. We will happy to add relevant content based on helpful comments from the reviewers. Revised and supplementary information including the mechanism of ferroptosis and the correlation between ferroptosis and HIBD is detailed on page 2

Results: How are driver, inhibitor/suppressor and marker genes defined?

Response: We apologize for the unclear definition of this part. This content has been revised and defined. We should explain what the Ferrb database does to help the reviewers understand. FerrDb database is the world's first manually curated database for ferroptosis regulators and ferroptosis-disease associations from published journal articles[2]. This kind of evidence is generally represented by an author statement of the role of the regulator in an original article[2]. At present, FerrDb database has been cited for 230 times, which is highly authoritative.

The database divides 259 ferroptosis-related genes into three categories based on confidence levels  and author statement : 108 drivers, 69 suppressors and 111 markers[2].

Driver: A gene that promotes ferroptosis.

Suppressor: A gene that prevents ferroptosis.

Marker: A gene that indicates ferroptosis occurrence

[2 ]Zhou N, Bao J. FerrDb: a manually curated resource for regulators and markers of ferroptosis and ferroptosis-disease associations. Database (Oxford). 2020 Jan 1;2020:baaa021.

Results: Stats have been performed on the graphed data in figure 5 but a description of these tests is missing from the methods or the figure legend.

Resonse: Thank you for pointing out the problem. We have briefly described the method in the revised manuscript.

Figures: the labelling on some of the figures is too small to be legible.

Response: Thank you for pointing out the issue. We have improved the size/quality of the label on Figure 7. See Figure 7 in the revised article for details.

Discussion

Define “UPR”

Response:Thank you for pointing out the issue. UPR:unfolded protein response. We have defined it in the discussion section of the revised article.

Round 2

Reviewer 1 Report

Th e manuscript has been revised. I have no additional comments.